# Adaptation of a Potyvirus Chimera Increases Its Virulence in a Compatible Host through Changes in HCPro

**DOI:** 10.3390/plants11172262

**Published:** 2022-08-30

**Authors:** Hao Sun, Francisco del Toro, Mongia Makki, Francisco Tenllado, Tomas Canto

**Affiliations:** 1Department of Microbial and Plant Biotechnology, Margarita Salas Center for Biological Research, Spanish National Research Council (CSIC), Ramiro de Maeztu 9, 28040 Madrid, Spain; 2Laboratory of Molecular Genetics, Immunology and Biotechnology, Faculty of Sciences, University of Tunis El Manar, Manar II, Tunis 2092, Tunisia

**Keywords:** HCPro suppressor of silencing, viral adaptation, potyvirus virulence, chimeric viruses, antiviral silencing suppression, viral infectious cycle, potyviral non-persistent transmission, viral host-range

## Abstract

A viral chimera in which the *P1-HCPro* bi-cistron of a plum pox virus construct (PPV-GFP) was replaced by that of potato virus Y (PVY) spread slowly systemically in *Nicotiana benthamiana* plants and accumulated to levels that were 5−10% those of parental PPV-GFP. We tested whether consecutive mechanical passages could increase its virulence, and found that after several passages, chimera titers rose and symptoms increased. We sequenced over half the genome of passaged chimera lineages infecting two plants. The regions sequenced were *5′NCR-P1-HCPro-P3*; *Vpg/NIa*; *GFP-CP*, because of being potential sites for mutations/deletions leading to adaptation. We found few substitutions, all non-synonymous: two in one chimera (nt 2053 *HCPro*, and 5733 *Vpg/NIa*), and three in the other (2359 *HCPro*, 5729 *Vpg/NIa*, 9466 *CP*). *HCPro* substitutions 2053 A**U**U(Ile)→A**C**U(Thr), and 2359 C**U**G(Leu)→C**G**G(Arg) occurred at positions where single nucleotide polymorphisms were observed in NGS libraries of sRNA reads from agroinfiltrated plants (generation 1). Remarkably, position 2053 was the only one in the sequenced protein-encoding genome in which polymorphisms were common to the four libraries, suggesting that selective pressure existed to alter that specific nucleotide, previous to any passage. Mutations 5729 and 5733 in the *Vpg* by contrast did not correlate with polymorphisms in generation 1 libraries. Reverse genetics showed that substitution 2053 alone increased several-fold viral local accumulation, speed of systemic spread, and systemic titers.

## 1. Introduction

HCPro is a non-structural, multifunctional dimeric protein of *Potyviruses* that among its functions performs two that are essential to the virus infection cycle: it suppresses gene silencing, neutralizing selectively plant antiviral defenses based on small RNAs (sRNAs) of viral sequence [1,2,3]. This facilitates the systemic infection of the plant and allows viruses to accumulate in systemic tissues. On the other hand, HCPro is mechanistically required to spread infection to other plants through aphid vectors [4,5]. There is a strong correlation between systemic viral titers in infected plants and the ability of insects to disperse potyviruses in a non-persistent way [6].

Since HCPro can determine viral titers in compatible infections, and those titers affect the ability of vectors to disperse the virus, it is possible that HCPro could act as a regulator that assures in the longer term that the virulence of viral isolates of different geographic origins and climates is adequate to allow optimal dispersal of infection within and between plants at the minimum possible cost for the virus, and thus the permanence of infection in such hosts and environments. HCPro could thus be playing a role in the adaptation of a virus strain to a particular host and environment condition.

Studies on the adaptation of natural potyvirus isolates to changing environmental conditions, to new hosts, or to both, found that the non-structural genome-linked protein Vpg from turnip mosaic virus (TuMV), accumulated alterations as the virus adapted to the new conditions [7]. This correlation suggests that highly variable regions within the Vpg play a significant role in adaptation. The Vpg is an intrinsically disordered protein that self-interacts, and also interacts with most other potyviral proteins, as well as with many host factors [8,9]. The Vpg is a second potyviral suppressor of RNA silencing [10], together with HCPro. The same studies on adaptation of a TuMV isolate naive to different arabidopsis accessions and under different environments through mechanical passages showed that the second protein accumulating mutations after the Vpg was HCPro, although at a distance from the former [7].

In contrast to the overall disorganized Vpg, HCPro has hypervariable regions limited to the protein N and C termini, and the number of sites under positive selection for the occurrence of non-synonymous substitutions that could help the virus adapt is lower than it would be expected randomly [11]. HCPro contains several functional domains that may not accumulate alterations easily, without having negative or even deleterious effects on essential functions of this protein, and thus on viral fitness. This limitation does not exclude that HCPro could acquire mutations that help the virus adapt.

In a previous study on how potyviral HCPro performs its silencing suppression, we had created a binary construct that expressed an infectious chimera [3] based on a plum pox virus (PPV) construct that had an added cistron that encoded green fluorescent protein (PPV-GFP) [12]. In the chimera, the *P1-HCPro* bi-cistron of PPV had been replaced by that of PVY. Viral titers of the chimera in infected *Nicotiana benthamiana* plants were low, around 5-to–10% of those of the parental PPV-GFP, and symptoms were very mild [3]. The artificial chimera was therefore more poorly adapted to this host than either of the parental viruses.

In this work, we assessed whether this infectious but artificial chimera, which lacks any nucleotide diversity, would yet possess the capability to adapt further to this compatible host increasing in virulence, and how this adaptation would relate to alterations in its genome. We found that the chimera could increase in virulence through mechanical passages. The two adapted chimeras analyzed had very few, and different, substitutions: combined, we found two in HCPro, two in the Vpg and one CP. We showed that mutations found in HCPro were likely induced under selective pressures that were occurring already in the agroinfiltrated plants (generation 1 viruses), previous to any mechanical passage bottleneck. Furthermore, we show that one single substitution in HCPro can alone increase virulence dramatically.

We have investigated additional properties of the chimera, relative to the parental viruses: its host range, and whether the chimera is transmitted by aphids.

## 2. Materials and Methods

### 2.1. Plants, Viruses and Plant Growing Conditions

Wild type (wt) *N. benthamiana* plants, as well as *N. benthamiana* transgenic plants that have their dicer-like *DCL 2* and *4* genes silenced (DCL2/4.5i plants) [13] were used in this study. *Nicotiana tabacum* cv. Xanthi nc and *Arabidopsis thaliana* ecotype Col 0 plants were also used.

*N. benthamiana* and tobacco plants were grown in growth chambers at 25 °C with a 16/8-h day/night photoperiod. Arabidopsis plants were grown at 21/18 °C and 16/8-h day/night photoperiod.

The potyviruses used in this study were: a Scottish PVY isolate (PVYO-SCRI-O; Genbank accession number AJ585196) [14]. Nucleotide annotations to its published *HCPro* cistron sequence are shown in [15]; a modified PPV derived from an infectious clone with an added cistron to express GFP (PPV-GFP) cloned in binary construct pLX-PPV [12], donated by Juan Antonio García (CNB, Cantoblanco, Madrid); two chimeric potyviruses in which the P1-HCPro bi-cistron of the Scottish PVY isolate replaced that of PPV-GFP [constructs pLX-(PVY P1-6x-HCPro)-PPV or pLX-(PVY P1-HCPro)-PPV] [3], expressing, respectively, chimeric viruses with (6x) or without a histidine tag (6xHis tag) fused to the N-terminus of PVY HCPro, respectively; a third chimeric construct was also used in which viral nt 2053 was altered (U→C), resulting in HCPro amino acid 352 substitution from Ile (A**U**U) to Thr (A**C**U). Cloning of this construct was performed as follows: a PCR fragment was amplified using template plasmid pLX (PVY P1-6x-HCPro)-PPV and oligo Fw: AGATTAATTAAGTACTAGTCCAG and Rv: ATGTCGCGAACTTTCTTTGTGAAATCCTTTGCATCCTCCTCGCTAATGTTA**G**TCAGCATT. The reverse oligo contains the nt 2053 (U→C) substitution (highlighted). The PCR fragment was digested with *Blp*I and *Nru*I, and cloned into construct pLX (PVY P1-6x-HCPro)-PPV linearized with the same enzymes, resulting in construct pLX (PVY P1-6x-HCPro_352Ile→Thr_)-PPV that expressed chimera 352Ile→Thr. The general structure of the parental virus, and of the three chimeric viruses is shown in Figure 1.

### 2.2. Infection of Plants with Viruses Using the Agroinfiltration Technique, or by Mechanical Inoculation

Plants (*N. benthamiana*, tobacco, arabidopsis) were infected with PPV-GFP or with the chimeric viruses using the infectious binary plasmids in which they were cloned. These binary plasmids were delivered into plant leaf tissues by agroinfiltration: in the case of *N. benthamiana* and tobacco plants, agrobacterium cell cultures carrying the binary constructs were grown exponentially at 28 °C in liquid LB media under shaking at 200 rpm, with the appropriate antibiotic selection (kanamycin resistance provided by the pROK2-based vectors, and rifampicin), pelleted, and re-suspended to an optical density at 600 nm of 0.3 in 2-N-morpholino ethanesulfonic acid (MES)-based infiltration buffer containing acetosyringone 0.2 mM to activate T-DNA transfer. After incubation at room temperature for 2 h, bacterial solutions were co-infiltrated together (1/1 *V/V* mixtures, each at OD 600 of 03). In the case of arabidopsis plants, the previous protocol, as well as the one described by [16], was used.

In mechanical passages of infections, *N. benthamiana* plants were also infected using extracts made from already infected plants: plant leaves were dusted with carborundum and inoculated mechanically with an ice-cold extract from an infected plant made with 0.1 M sodium phosphate buffer, pH 8 (1/1 weight/volume). The extract (25-to−50 μL/leaf) was rubbed lightly on the leaf surface with gloved hands, and the leaves were washed with distilled water immediately afterwards.

### 2.3. Assessment of Viral Infections

The presence of local or systemic viral infections, as well as a measure of relative viral titers, was made in two ways:

(a) Measuring viral protein levels in infected plant tissues by western blot, using anti-PPV coat protein (CP) and anti-GFP antibodies: leaf discs were mechanically ground in extraction buffer (0.1 M Tris-HCl PH 8, 10 mM EDTA, 0.1 M LiCl, 1% β-mercaptoethanol and 1% SDS), boiled for 5 min, clarified by bench centrifugation and fractionated in 15% SDS-PAGE gels, and transferred by western blot to PVDF membranes. To detect and quantify the viral CP, a rabbit polyclonal antiserum to the PPV CP was used (Loewe Biochemica, Sauerlach, Germany). A rabbit GFP polyclonal antiserum donated by G. Cowan and L. Torrance (The James Hutton Institute, Dundee, UK) was also used. A commercial alkaline phosphatase-labeled goat anti-rabbit secondary antibody (Sigma-Aldrich, St. Louis, MO, USA) and BCIP/NBT substrate solutions (Duchefa, Haarlem, The Netherlands) were also used. Western blot membranes were scanned, and relative band densities were quantified using Image J software (www.imagej.nih.gov, accessed on 10 July 2022).

(b) Measuring viral genomic RNA levels by RT-qPCR using one-step reverse transcription plus real-time quantitative polymerase chain reactions (RT-qPCR). Briefly, total RNA was extracted from leaf tissue using TRIzol reagent (Invitrogen, Waltham, MA, USA) following the manufacturer’s instructions, and contaminant DNA was removed by treatment with TURBO DNA-free kit (Ambion, Austin, TX, USA). RT-qPCR was performed using a final reaction volume of 15 uL that contained 7.5 μL of Brilliant III Ultra-Fast RT-qPCR Master Mix (Agilent, Santa Clara, CA, USA), 1.8 μL of RNase-free water, 0.75 μL of reverse transcriptase (Agilent), 0.15 μL of 100 mM dithiothreitol (Agilent), 0.3 μM each primer, and 3 μL of total RNA extract (approximately 15 ng RNA/μL). Relative quantifications were calculated by the ΔΔCt method. 

A pair of oligos of PPV sequence was specifically designed and tested for the qPCR amplification of PPV-GFP and of the chimera: qPCR PPV Fw: AAGTCGATGGGCGAACTATG, and qPCR PPV Rv: AAACCGAAGTCCACAACCAC. In addition, oligos 18SrRNA-Fw (5′-GCCCGTTGCTGCGATGATTC-3′) and 18SrRNA-Rv (5′- GCTGCCTTCCTTGGATGTGG-3′) for 18S rRNA amplification were used for normalization. RT-qPCRs were performed in a Rotor-Gene Q thermal cycler (Qiagen, Hilden, Germany) using the following thermal protocol: 50 °C for 10 min; 95 °C for 3 min; 40 cycles of 95 °C for 10 s and 60 °C for 20 s; and a final ramp for melting analysis from 60 °C to 95 °C rising 1 °C every 5 s.

Results from either the serological quantitation of CP/GFP viral protein levels or from the RT-qPCR quantitation of viral genome levels were analyzed with SPSS STATISTICS (IBM, Armonk, NY, USA).

In addition to measuring viral titers, the virulence of infections in *N. benthamiana* plants was assessed visually, looking for the presence of chlorotic mosaic and leaf curling (typical PPV symptoms in this host), relative to non-infected plants. 

### 2.4. Transmission Assays with Aphids

Aphid transmission assays were performed using a clone of the peach aphid *Myzus persicae* Sulz. Systemic *N. benthamiana* fully expanded leaves infected with the virus were used as donors to feed aphids that had been starved for the previous 2 h. Detached leaves were washed with mild detergent to remove the aphid repellants present in this plant species, rinsed with water and gently dried. Aphids fed on the abaxial side of the leaves placed inside wet glass container chambers. Feeding time was ~15 min, after which the insects were individually transferred to small, healthy *N. benthamiana* plantlets. Each plantlet received 10 aphids. Insects were killed 24 h after the transmission assay, and plants were allowed to grow for 20 days. At that time, leaf samples were taken, and the presence of the virus was determined by western blot with an antibody against the CP of PPV, as already described.

### 2.5. Analysis of sRNA Populations Present in Systemic Tissue of Plants Infected with the Chimeras

Data from four next-generation sequencing (NGS) libraries of sRNA populations present in systemically infected tissues of *N. benthamiana* plants that had been agroinfiltrated with viral chimeras were available to us [3]. These high-throughput sequencing data are deposited and are accessible at the Gene Expression Omnibus (GEO, http://www.ncbi.nlm.nih.gov/geo; accession no. GSE135651, accessed on 10 July 2022). Using MISIS-2 [17] and our own Perl scripts, we identified single nucleotide polymorphisms (SNPs) in reads of sRNAs that corresponded to the viral genome sequence. To be considered, an SNP would have to have at least 100 reads, and the nt of reference would have to be detected in less than 95% of the reads in at least one of the four libraries analyzed.

### 2.6. Analysis of Genomic Regions of the Chimeras after Several Mechanical Passages in N. benthamiana Plants

Three genomic regions of two viral chimeras present in two different plants after they had undergone several consecutive mechanical passages (generations) through *N. benthamiana* plants were amplified by RT-PCR from total RNAs extracted from those two plants, to sequence them and to look for nucleotide changes/deletions, relative to the original sequence. Appropriate primers were used (Table 1). They produced fragment 1 (2403 nt in length), fragment 2 (900 nt), and fragment 3 (1929 nt). Fragment 1 amplified the *P1-HCPro* bi-cistron + flanking areas; Fragment 2 amplified the *Vpg* cistron + flanking areas; Fragment 3 amplified the *GFP* and *CP* cistrons + flanking areas. Together they covered 5232 nt, just over half of the 10,452 nt of the (PVY P1-HCPro)-PPV-GFP genome (without the polyA tail).

To amplify these three fragments, cDNAs were generated from total RNAs extracted from three discs obtained from two systemic leaves of each of the two infected *N. benthamiana* plants, using SuperscriptIII DNA polymerase, following the manufacturer’s instructions (Invitrogen). As reverse oligos primers D, F and B were used (Table 1). Amplification by PCR of the three fragments from the cDNAs was performed using high-fidelity Phusion DNA polymerase (Invitrogen): fragment 1 (primers A and D), fragment 2 (primers E and F), fragment 3 (primers H and B). PCR amplification conditions were [98 °C 1 min] 1 cycle; [98 °C 10 sc/50 °C 10 sc/72 °C 25 sc] 35 cycles. In addition to these three genomic coding regions, we amplified by RT-PCR in the same way the 5′ untranslated region (UTR) of the two viral chimeras, which is limited to 35 nt in the cloned virus, plus the start of the P1 (146 nt in total) using primers A′ and C′ (Table 1).

Amplified PCR fragments were agarose gel-fractionated, excised and extracted with QIAEX DNA gel extraction kit (Qiagen). Cleaned fragments were then sequenced (Secugen SA). Contigs of the sequence reads were assembled into the three fragments and compared to the original clone sequence using Vector NTI software (Thermofisher Scientific/Invitrogen). 

## 3. Results

### 3.1. Changes in Systemic Titers of Viral Chimeras after Consecutive Passages (Generation 1 vs. 4, 5)

We tested if the infectious chimeras would have the ability to adapt further to *N. benthamiana* through consecutive mechanical passages. To do this, extracts from systemically infected leaves from plants agroinfiltrated with constructs pLX-(PVY P1-6x-HCPro)-PPV or pLX-(PVY P1-HCPro)-PPV (generation 1 chimeras) were used to inoculate mechanically a new set of healthy plants (creating generation 2 chimeras). Successive mechanical passages were made as indicated in Figure 2a. Each generation lasted ~30 days. Inoculated plants were wt or DCL2/4.5i plants that have the ability to create primary sRNAs of 21 and 22 nt against viral sequences compromised [13], as indicated (Figure 2a). After those passages, two infected plants, A and D, were selected (Figure 2a, generations 3 and 4, respectively, highlighted in green). With extracts made from each of the two plants both, wt and DCL2/4.5i plants were inoculated (generations 4 and 5, respectively). Systemic viral titers in the inoculated plants were measured at 16 dai and were found to be significantly higher than those occurring in agroinfiltrated (generation 1) wt plants (Figure 2b). Interestingly, the passaged chimeras in generations 4 and 5 plants failed to accumulate to higher levels in DCL2/4.5i than in wt plants (Figure 2b). In this regard, we had previously observed that systemic chimera titers (generation 1) were significantly higher in DCL2/4.5i plants than in wt plants while those of PPV-GFP were lower [3]. Thus, the generation 4−5 adapted chimeras originating from plants A and D showed relative accumulation patterns in wt vs. DCL2/4.5i plants that were similar to that of the parental PPV-GFP, rather than to that of generation 1 chimeras.

### 3.2. Genomic Changes in the Chimeras after Consecutive Mechanical Passages (Generations 3, 4)

To study how the increases in generation 4 and 5 chimera titers related to alterations in their genomes we amplified by RT-PCR fragments of the genomes of the chimeras infecting plants A and D. Three fragments were amplified that covered more than 5200 nt, over half of the viral genome (Figure 3a). The RT-PCR fragments amplified from the two plants were sequenced. Contigs were assembled and compared with the chimera sequence encoded by construct pLX-(PVY P1-HCPro)-PPV. Surprisingly, no synonymous substitutions were found. No deletions were found either, and the non-viral *GFP* cistron was kept by both the chimeras in plants A and D. Very few non-synonymous substitutions that resulted in amino acid changes were found: only two in the virus infecting plant A, and three in the virus infecting plant D, and they were different. These single substitutions in the polyprotein coding region affected the *HCPro* and *Vpg/NIa* cistrons in the chimera from plant A, and the *HCPro, Vpg/NIa* and *CP* cistrons in the chimera from plant D (Figure 3b). An additional extra nt was found in the sequenced part of the 5′end non-coding-region (NCR) of plant A (an adenine, after nt 110).

With regard to the two substitutions affecting *HCPro*, the one at nucleotide 2053 in the virus infecting plant A changed HCPro amino acid 352, in the C-terminal third of the protein: Ile (A**U**U)→Thr (A**C**U). This is a non-conservative substitution of a non-polar (Ile) residue by a polar (Thr) one. The other substitution at nucleotide 2359 in the chimera infecting plant D led to a substitution at HCPro amino acid 454, Leu (C**U**G)→Arg (C**G**G). This substitution restored the intact sequence of PVY HCPro near its cleavage site from P3 (Figure 4), which had been altered during the cloning of chimera constructs because keeping a unique *SexA*I site in the *PPV-GFP sequence* was required [3].

We found a single mutation in the *Vpg/NIa* cistrons in each of the chimeras infecting plants A and D, respectively. They did not coincide but were only 2 nt apart: at nucleotide 5733, which led to an Asp (**G**AC)→Tyr (**U**AC) substitution in the chimera infecting plant A, and at nucleotide 5729, leading to a Met (AU**G**)→Ile (AU**A**) substitution in the chimera infecting plant D. We also found a single substitution in the *CP* cistron of the chimera in plant D, at nt 9466, in the N-terminal third of the protein (Figure 3b).

### 3.3. Analysis of Single Nucleotide Polymorphisms in sRNA Populations of Viral Sequence in Systemic Tissues of Agroinfiltrated (Generation 1) Plants

Four separate NGS libraries of sRNA reads were available from infected systemic tissues of *N. benthamiana* plants that had been agroinfiltrated with the binary constructs pLX-(PVY-P1-6x-HCPro)-PPV or pLX-(PVY-P1-HCPro)-PPV, expressing the viral chimeras [3]. Deep sequencing analysis of the sRNA reads of viral sequence present in those plants infected with generation 1 chimeras showed the presence of SNPs that distributed throughout the whole viral genome, both in non-coding and in coding regions (120 SNPs detected at least in one of the four libraries; Figure 5a). Thus, sites of genetic variability were generated from the single agroinfiltrated viral chimeric sequence during the initial infection. In some nucleotide sites, polymorphisms were found in more than one library. However, only in four sites (3 in the *5′UTR*: nt 24, 28, and 35, and one in the *HCPro* cistron, nt 2053) the polymorphisms were common to the 4 libraries (Figure 5b).

The two substitutions in HCPro found in the sequenced regions of the two chimeras infecting plants A and D coincided with SNPs detected in the libraries from generation 1 infected plants. Interestingly, the substitution in the HCPro encoded by the chimera infecting plant A corresponded to the SNP in the *HCPro* cistron that was found in all four libraries, which was the only one along the protein-encoding genome with that property (Figure 5b). The second amino acid substitution, in the HCPro encoded by the chimera infecting plant D corresponded to a substitution in *HCPro* found in one library. None of the other three amino acid substitutions detected (in *Vpg/NIa* and *CP*) coincided with any of the SNPs found in the generation 1 sRNA libraries.

### 3.4. The Effect of HCPro Nucleotide Substitution 2053 on the Virulence of the Chimera

We investigated the effect on titers and symptoms of this single nucleotide 2053 substitution (HCPro amino acid 352 Ile→Thr substitution) that was appearing recurrently among reads of viral sRNAs induced by generation 1 chimera infections, and which had been incorporated in the chimera infecting plant A. To do this we compared the virulence in *N. benthamiana* plants agroinfiltrated with constructs pLX-(PVY P1-6x-HCPro)-PPV vs. pLX-(PVY P1-6x-HCPro_532Ile→Thr_)-PPV, expressing the unmodified chimera, or chimera 352 Ile→Thr. Systemic infection symptoms were detected as early as at 8 dpi in the case of all plants infected with the latter chimera, independently of whether they were wt or DCL2/4.5i plants. Symptoms consisted of mosaic and slight leaf distortion (not shown), By contrast, no symptoms were apparent at 8 dpi in the case of the unmodified chimera. To test how the original chimera vs. chimera 352 Ile→Thr accumulated locally we compared CP titers in leaf discs agroinfiltrated within the same leaf, at 5 dai. Accumulation of chimera 352 Ile→Thr in agroinfiltrated tissues was much larger than that of the original chimera in either wt or in DCL 2/4.5i plants (Figure 6a). In wt plants, serological detection of viral CP levels of the original chimera was so low that we could not make an accurate statistic estimate of the fold-change in titers, probably because of their being below the threshold of sensitivity of the antibody. However, RT-qPCR analysis of viral genomic RNA levels from the same tissues indicates that the difference in genome levels was ~4-fold (Appendix A). We then compared the presence of the two chimeras in systemic leaf tissues, at 10 and at 20 dpi. At 10 dpi, the unmodified chimera was still undetectable in systemic leaves of most plants, whereas chimera 352 Ile→Thr was already present in all plants, indicating a faster spread of the latter. At 20 dpi both chimeras were present in systemic tissues, although the unmodified chimera had much lower titers (Figure 6b). These data indicate that amino acid substitution 352 Ile→Thr at HCPro is alone sufficient to increase the local accumulation, speed of systemic spread, and systemic titers of the viral chimera in this host.

### 3.5. Host Range and Transmissibility of the Chimeric Virus 

PVY infects systemically *N. benthamiana* and tobacco plants, but not arabidopsis plants, whereas PPV infects systemically *N. benthamiana* and arabidopsis plants, but not tobacco plants, although the virus is able to replicate in tobacco cells [19]. We tested the host range of the chimeras relative to those of the parental virus. We agroinfiltrated tobacco plants with binary plasmids that expressed either the unmodified 6x his-tagged chimera or chimera 352 Ile→Thr. We tested by western blot the presence of viral CP and GFP in agroinfiltrated leaf tissues at 5 dai, and found that both chimeras accumulated locally, the latter to higher levels (Figure 7).

With regard to arabidopsis plants, we could detect some faint accumulation of CP and GFP bands only in some of the leaves infiltrated with the construct that expresses chimera 352 Ile→Thr (Figure 7). We failed to detect any of the two chimeras in systemic tobacco tissues at 22 and 37 dai or arabidopsis leaves at 20 dai (not shown). Therefore, with regard to the infection of these three hosts, the chimera range is not additive to those of the parental viruses, but more restrictive (Figure 7).

We tested whether the unmodified chimeras (with or without 6x His tag), as well as chimera 352 Ile→Thr could be transmitted by the peach aphid *Myzus persicae* from systemic leaves of infected *N. benthamiana* plants. In four independent transmission experiments (two experiments involving donor plants infected with the unmodified chimera with 6x His tag, and two involving plants infected with the chimera 352 Ile→Thr) we could not achieve transmission. In experiments involving 4 chimera-infected donor leaves, 280 aphids and 28 recipient plantlets (10 aphids/plantlet) there was 0 transmission of infection; the positive control of a PVY-infected leaf, 5 recipient plantlets and 50 aphids (10 aphids/plantlet) gave 4 out of 5 of the recipient plantlets infected). Therefore, the chimeras in their current form are either non-transmissible or are transmitted very poorly by aphids.

## 4. Discussion

In our previous studies on two functions of HCPro (silencing suppression inside the plant, and mediation of insect transmission between plants) we created mutant and tagged forms of PVY HCPro that we expressed in plants either from binary vectors in a virus-free context or from a heterologous potato virus X (PVX) vector [2,4,15,20,21] since we lacked an infectious clone of PVY amenable to manipulation in our hands. Recently we created infectious potyviral chimeras based on a PPV-GFP that had its *P1-HCPro* bi-cistron replaced by that of PVY (Figure 1) and [3]. This allows us to express PVY HCPro mutants and variants in the context of a potyviral infection. However, the chimeras spread and accumulate poorly (Figure 2b) and [3], and this fact handicaps their use in research. We had observed that in DCL2/4.5i plants with their silencing suppression mechanisms compromised [13] the chimeras accumulated more, suggesting that the antiviral silencing machinery was contributing to the chimera’s reduced virulence (viral titers and symptoms), whereas this effect of these plants was not observed for the parental PPV-GFP [3]. The reason for the latter is not clear.

On the other hand, how plant viruses adapt and evolve in response to new hosts and environments, and vectors, is currently a very interesting topic of research, as climate change and increased global trade multiply the opportunities for those situations to occur [22]. Studies on *Potyviruses* indicate that mutations leading to adaptation do not distribute uniformly along the viral genome but target instead specific variable regions of some of the encoded viral proteins [7,11].

We wanted therefore to test whether we could increase the virulence of our chimera for practical purposes, but also to study how an un-adapted virus could evolve towards becoming better adapted to its host. The fundamentals of adaptation of natural viruses to new contexts may or may not differ from those of adaptation to a compatible experimental host (*N. benthamiana*) of an intrinsically un-adapted, artificial virus lacking any starting sequence diversity.

We agroinfiltrated plants with the chimera constructs (generation 1 plants) and after several consecutive mechanical passages that lasted around four months (Figure 2a) we selected two plants, A and D (generations 3 and 4, respectively) that we used to inoculate either wt or DCL2/4.5i plants (generations 4 and 5). We analyzed systemic viral titers in these inoculated plants and found that they were several-fold higher than those of generation 1 plants. Interestingly, these “adapted chimeras” did not accumulate to higher levels in DCL2/4.5i plants than in wt plants (Figure 2b). In this aspect, they now behaved like the parental PPV-GFP, rather than like the original chimeras.

We sequenced three fragments covering together over half of the genomes of the chimeras infecting plants A and D (Figure 3a). The reasons for selecting those regions are that the *P1HCPro* bi-cistron (in fragment 1) is a heterologous sequence that originates from another potyvirus, PVY; that the *Vpg-NIa* (in fragment 2) contains hyper-variable regions (also *HCPro*), where non-synonymous substitutions have been shown to occur under positive selection, facilitating the genetic adaptation of viral isolates [7,11]; and for fragment 3, there are two reasons, the first one is that the *GFP* cistron is a non-viral sequence that could be lost, and the second one that the *CP* has been also reported as a potential target for viral adaptation and evolution [11,23].

We found that the heterologous *GFP* cistron had been maintained through the months of passages, suggesting that its fitness cost to the virus in this host is low, or lower than the cost of deleting it. Overall, we could find only a few non-synonymous substitutions, but different among the two chimeras, indicating that divergent chimera evolution in different plants had occurred (Figure 3b) that led both to increased titers. Amino acid mutations were two in the chimera infecting plant A (in the C-terminal third of HCPro, and the Vpg) and three in the other one (at the C-terminus of HCPro, the Vpg and the N-terminal third of the CP). Interestingly, they all fell within proteins or protein domains that have been associated with potential sites for mutations leading to adaptation, as mentioned. The mutation in amino acid 352 of HCPro encoded by the chimera infecting plant A is non-conservative (Ile→Thr). On the other hand, the mutation in HCPro found in the chimera infecting plant D restored the exact sequence of PVY HCPro (Figure 4). The *Vpg* mutations in the chimeras infecting plants A and D were not the same but were only two nucleotides apart. When aligning with TuMV YC5, the amino acid changes in HCPro and in the Vpg found in the chimeras infecting plants A and D did not correspond with mutations in the TuMV isolate after adapting to different arabidopsis accessions [7].

Four NGS libraries of reads of sRNA populations from systemic leaves of wt plants infected with generation 1 chimeras were available to us. We looked for the presence of SNPs in those sRNA reads, and found 120 along the viral genome (Figure 5), including the two nucleotide positions that cause the mutations in the HCPro of the chimeras infecting plants A and D. Remarkably, mutation at nucleotide 2053 that causes HCPro amino acid 352 Ile→Thr substitution in the chimera infecting plant A was found in the four libraries. It is the only position along the whole protein-encoding genome with that property. On the other hand, mutations in the Vpg of the chimeras infecting plants A and D, or in the CP of the chimera infecting plant D did not correlate with SNPs found in any of the four libraries from generation 1 infections.

These data indicate that mutations in specific sites of the viral genome are occurring repeatedly, prior to any passage bottlenecks, and that selective pressures are already operating in generation 1 infections that will lead to some becoming eventually prevalent, as happened to those in positions 2053 and 2359 in the *HCPro* cistron in the chimeras infecting plants A and D. Generation 1 viral genomic RNAs with the mutations in *HCPro* may perhaps be advantaged in their replication/accumulation, and/or local or systemic spread (the libraries originate from systemically infected tissues). In the case of the mutation in HCPro of the chimera infecting plant D, restoration of the intact PVY HCPro sequence may have led to a slightly more efficient polyprotein processing, although western blot data shows that the original unmodified chimera processes HCPro properly (not shown). In the case of mutation of HCPro amino acid 352, we have no indication of why it could be selected.

To test the effect on virulence of the nucleotide 2053 mutation leading to HCPro amino acid substitution 352 Ile→Thr, we incorporated it into the original chimera, creating chimera 352Ile→Thr, and compared both viruses. We found that chimera 352Ile→Thr titers were several-fold higher in agroinfiltrated tissues as well as in systemic tissues than those of the unmodified chimera, and also that the virus moved much faster (Figure 6). Therefore, a single nucleotide substitution in HCPro dramatically increased the virulence of the chimera.

We finally tested the host range of our chimera relative to those of the parental viruses. PVY infects many solanaceous plants of economic importance such as potato, pepper or tomato. PPV has also devastating effects on bone fruit trees (*Prunus*). With regard to experimental plants, PVY infects *N. benthamiana* and tobacco, but not arabidopsis Col 0 plants. PPV-GFP by contrast infects *N. benthamiana*, arabidopsis Col 0, and the agroinfiltrated leaves of tobacco leaves but does not spread systemically [20]. We found that the host range of the chimeras is more limited than that of each parental virus (Figure 7). We tested the transmissibility by aphids of the chimeras but could not transmit them, even chimera 352Ile→Thr that had higher titers. This was somewhat unexpected, as it is known that PVY HCPro can mediate the transmission of PPV in trans and *in vitro* [24], and the CP of PPV-GFP and the chimeras contain the DAG motif. Why transmission failed in cis and *in vivo* is a matter for further research.

In conclusion, we show that an un-adapted artificial chimeric potyvirus lacking any initial sequence variability can adapt to a compatible host and increase in virulence through mutations in its genome. We found that different mutations can lead to increased virulence in different chimera lineages (plants A vs. D). We show that in generation 1 infections, selective pressures created genomic variability at specific nucleotides, which with regard at least to those in *HCPro* would lead eventually to their incorporation prevalence in later passages. In the case of the single non-conservative amino acid substitution in HCPro, 352 Ile→Thr, we demonstrate that it can dramatically increase the virulence of the chimera. This happens in spite of this substitution not restoring any natural HCPro sequence known to us, as remarkably, the Thr at this position seems to be not found among natural potyviruses (Appendix A). This could suggest that this novel substitution is caused by the chimeric nature of the virus, perhaps then related to the need for interactions between HCPro (from PVY) and other viral components (from PPV). We also do not know how this substitution affects specific molecular properties of the protein and its functions. For example, does it increase or diminish its suppression activity? [25]; or does it affect dimerization, or the ability of dimers to relocate to the microtubules in response to osmotic stress? [21], but all this will be a matter for our future research.

## Figures and Tables

**Figure 1 plants-11-02262-f001:**
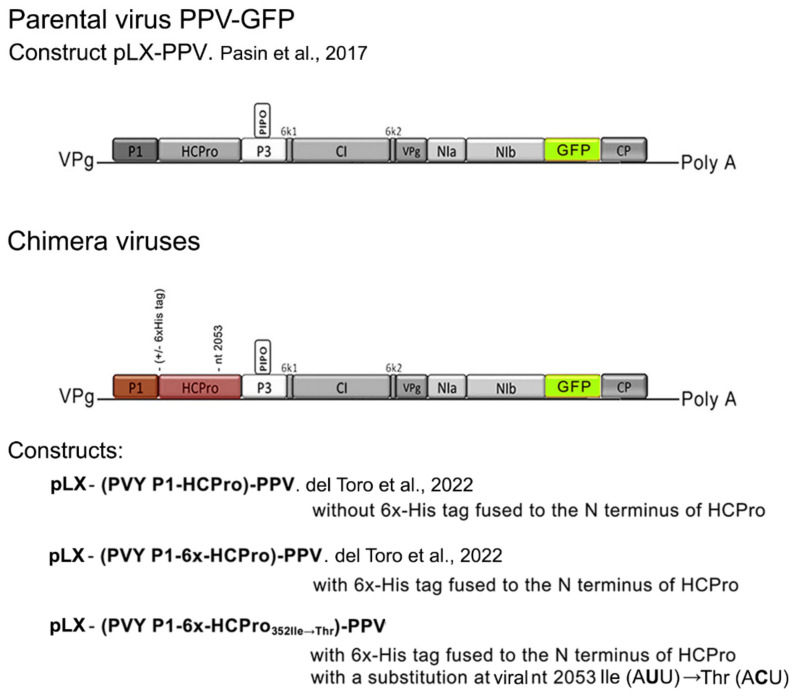
Schematic representation of the genomic organization of the viruses used in this study. A plum pox virus (PPV)-based construct encoding a GFP-expressing infectious PPV variant, PPV-GFP (PLX-PPV; Pasin et al., 2017), and three constructs expressing chimeric viruses, in which the *P1-HCPro* bi-cistron of PPV had been replaced by that of a Scottish isolate of Potato virus Y (PVY), shown in red. In two chimeric constructs, the PVY HCPro was expressed either with or without a 6x Histidine tag fused at its N-terminus (chimera with or without 6x His tag). A third construct expressed a viral chimera in which nt 2053 was modified, expressing a 6x His-tagged HCPro with a single substitution at amino acid 352: Ile (A**U**U) →Thr A**C**U) (chimera 2053Ile→Thr). All virus variants carry an additional cistron that expresses GFP.

**Figure 2 plants-11-02262-f002:**
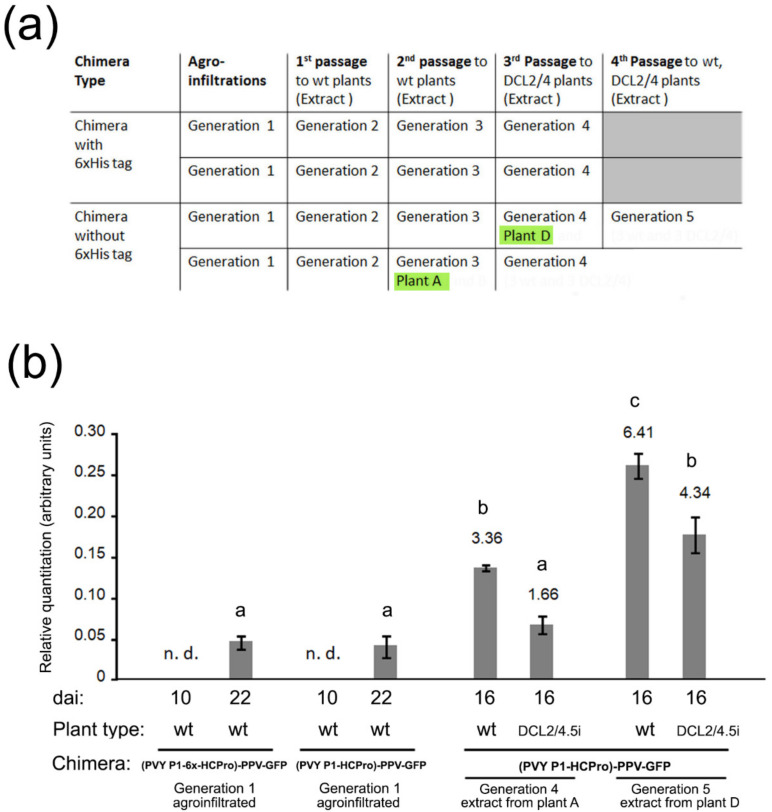
**Changes in chimera titers after several consecutive mechanical passages through *Nicotiana benthamiana* plants:** (**a**) Schematic representation of the procedure followed: after agroinfiltration of wild type (wt) plants with binary constructs pLX-(PVY P1-6x-HCPro)-PPV or pLX-(PVY P1-HCPro)-PPV, expressing infectious chimeras (generation 1), extracts from these infected plants were used to mechanically inoculate healthy plants (generation 2), which were used in turn to inoculate the next round of plants, creating generations 3→4→5, successively. Two Infected plants A and D that were subsequently used to sequence the genomes of the chimera viruses infecting them are highlighted in green; (**b**) RT-qPCR quantitation of relative systemic titers of genomic viral RNAs of the chimeras found in plants inoculated with extracts from plants A and D (generation 4 and 5 chimeras) vs. those found in generation 1 agroinfiltrated plants. They are all relativized to the levels of PPV-GFP at 7 dai (value 1). After the mechanical passages, chimera titers in systemic plant tissues increased significantly in wt plants. Titers of the passaged chimeras were significantly lower in DCL2/4.5i plants than in wt plants. Different letters indicate significant differences (*p*-value < 0.05; Dunett’s T3 test).

**Figure 3 plants-11-02262-f003:**
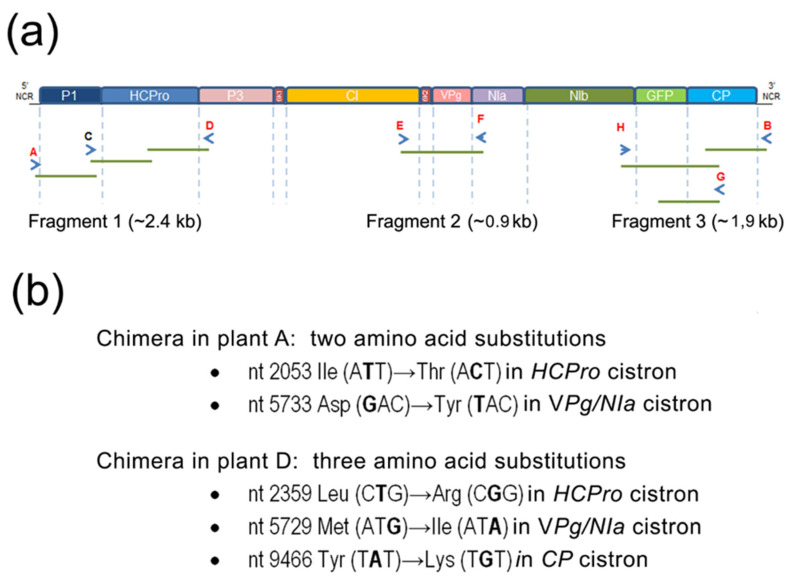
Characterization of genomic alterations in the evolved chimeras present in plants A and D: (**a**) Schematic representation of the three viral genomic regions (fragments 1, 2, 3 that together account for more than half of the viral genome) amplified by RT-PCR from infected plants A and D (the same plants indicated in Figure 2) and sequenced. The primers used are specified in Table 1; (**b**) Results of the sequencing of the PCR fragments 1, 2 and 3 of viral sequence obtained from each of the two plants. They indicate that the chimera in plant A had two confirmed amino acid substitutions, relative to the agroinfiltrated generation 1 chimera, while that in plant D had three substitutions. The five substitutions were different.

**Figure 4 plants-11-02262-f004:**
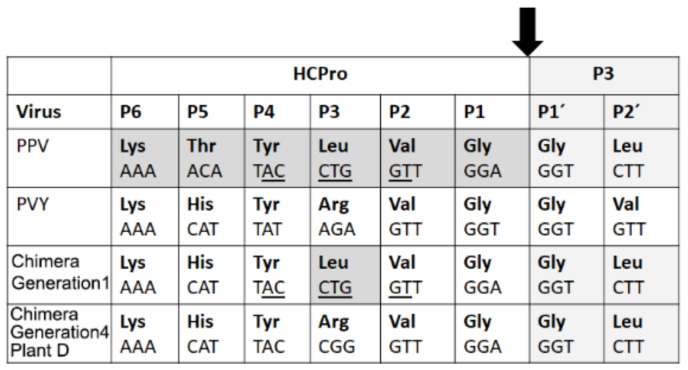
Amino acid sequence and substitutions at the C-terminal HCPro/P3 cleavage sites of the parental viruses and the chimeras, as described in [18]. A single Arg to Leu substitution (position P3) in the HCPro expressed by the chimeras differentiates it from the parental PVY HCPro because of the use during the cloning of the binary constructs encoding them maintaining of a *SexA*I site present in PLX-PPV-GFP (ACCWGGT. W: A or T) was required. However, RT-PCR showed that in the generation 4 chimera infecting plant D nucleotide substitution 2359 led to HCPro amino acid 454 substitution Leu (C**T**G)→Arg (C**G**G), which restored the intact amino acid sequence of PVY HCPro in this chimera, although not its nucleotide sequence. DNA sequences are shown. P′1 and P′2 indicate the 1st and 2nd amino acid positions at the P3 viral protein.

**Figure 5 plants-11-02262-f005:**
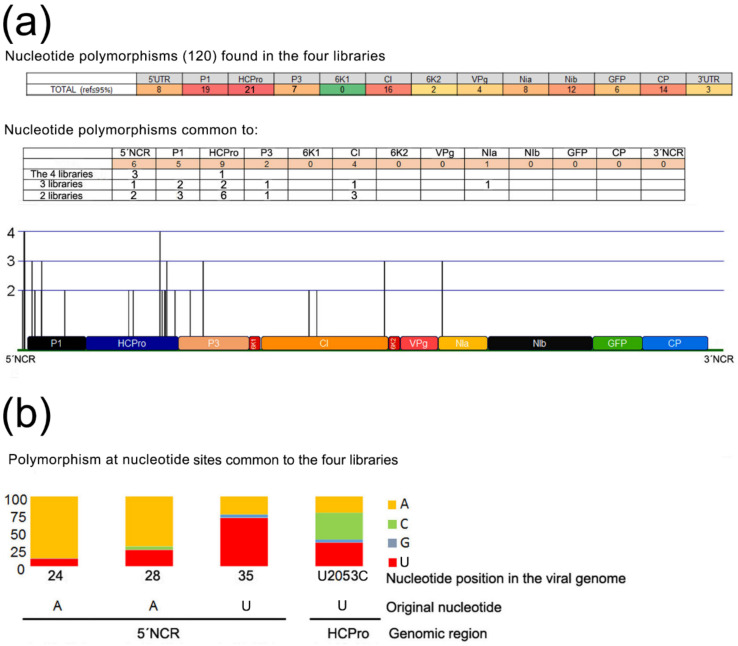
Nucleotide variability in agroinfiltrated, generation 1 plants systemically infected with chimeras. (**a**) Detected in four separate deep sequencing libraries of viral small RNA reads: two from plants infected with (PVY P1-HCPro)-PPV-GFP, and two from plant infected with (PVY P1-6x-HCPro)-PPV-GFP. Single nucleotide polymorphisms common to the 4, 3 or 2 libraries, as well as their occurrence along the viral non-coding-regions (NCRs) and cistrons are indicated. (**b**) The four nucleotide polymorphisms common to the 4 libraries were located in the *5′NCR* and in the *HCPro* cistron.

**Figure 6 plants-11-02262-f006:**
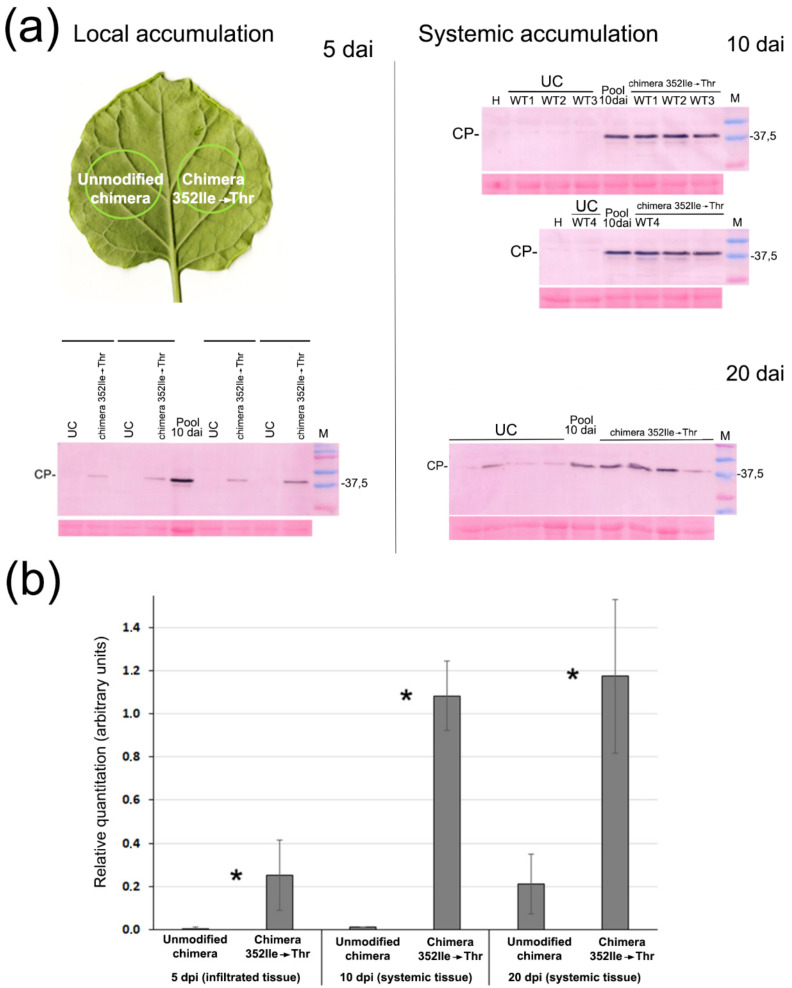
The effect on chimera titers of substitution Ile (AUU)→Thr (ACU) in amino acid 352 of HCPro in wt *N. benthamiana* plants: (**a**) Western blot analysis of coat protein (CP) levels to assess the relative accumulation of the unmodified chimera vs. chimera 352Ile→Thr locally in infiltrated discs (within the same leaf) at 5 dai, and systemically at 10 and 20 dai. Accumulation of chimera 352Ile→Thr was significantly higher. Lanes labelled UC, unmodified chimera; lanes labelled pool are loaded with a sample that is an equal mix of extracts of the wt plants infected with chimera 352Ile→Thr at 10 dai, used as control for comparisons between blots. H, healthy plant extract; M, markers of molecular size, in kDa. Panels below each blot are the membranes stained with ponceau S (rubisco band) as controls of loading; (**b**) Chart of relative accumulation of the unmodified chimera vs. chimera 352Ile→Thr. Pool of wt plants infected with chimera 352Ile→Thr at 10 dai, value 1. Statistical analysis of titers at 5 and 10 dpi employed Student’s T-test since data showed Gaussian distribution. At 20 dpi analysis employed Mann–Whitney U test, since data did not show Gaussian distribution. Asterisks indicate significant differences between treatments (* *p*-value < 0.05; Student *t*-test; Mann-Whitney).

**Figure 7 plants-11-02262-f007:**
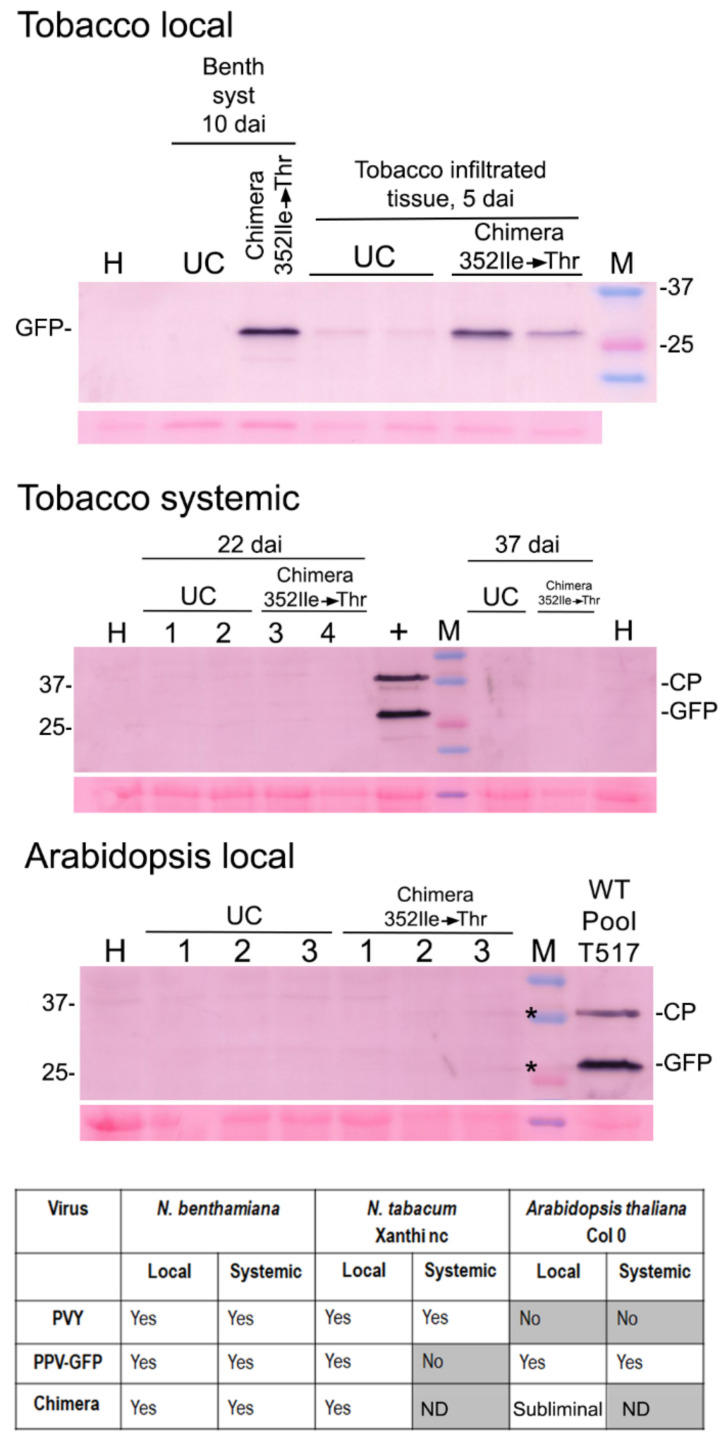
Assessment of the chimera host range relative to the parental viruses. The ability of the chimera to infect systemically three plant species. Upper blot shows western blots used to detect the coat protein (CP) of PPV and/or the GFP in agroinfiltrated or systemic tissues. Results indicate that both, the unmodified chimera (UC) and chimera 352Ile→Thr accumulated locally, but not systemically. H, healthy plant extract; M, markers of molecular size, in kDa. +, positive control of *N. benthamiana* infected with chimera 352Ile→Thr. Asterisks are placed adjacent to two faint bands corresponding to the CP and GFP expressed by chimera 352Ile→Thr, detected in sample 3 of infiltrated Arabidopsis leaf tissue. Panels below each blot are the membranes stained with ponceau S (rubisco band) as controls of loading. The scheme below the blots summarizes the overall results. Yes/No means viral presence/absence (shown by us, or by others in previous publications), respectively. ND, means not detected by us in these experiments.

**Table 1 plants-11-02262-t001:** The oligonucleotides used to amplify by RT-PCR the three segments of the chimera genome.

Oligo Name	Sequence	Viral Region	nt Position in Viral Chimera Sequence in Construct pLX-(PVY P1-HCPro)-PPV	Polarity
A	caatcaaatcaatctcaagc	5′ UTR	39–58	+
A′	aaaatataaaaactcaacac	5′ UTR	1–20	+
C	tggtccaatcaagtccgcac	P1	807–826	+
C′	cttgacttgcagtaaatttgg	5′ UTR-P1	126–146	−
D	gatttgtggcttgtaaatgc	P3	2422–2441	−
E	tgctagaattcaagaacctg	CI	5392–5412	+
F	cagatgagttattcaattgg	NIa	6272–6291	−
G	atagtttcgaagtcattgcc	CP	9575–9594	−
H	ggagacagcactgaagaagc	NIb	8375–8394	+
B	aggaatctaaaaacaactgg	3′ UTR	10,284–10,303	−

## Data Availability

NGS libraries of sRNA reads are available at the Gene Expression Omnibus (GEO, http://www.ncbi.nlm.nih.gov/geo; accession no. GSE135651, accessed on 10 July 2022).

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
