# Peer review of "Adaptation of a Potyvirus Chimera Increases Its Virulence in a Compatible Host through Changes in HCPro"

_plants, 2022, doi:10.3390/plants11172262_

Round 1

Reviewer 1 Report

In this manuscript the adaptation of a chimeric potyvirus PVY+PPV is studied. After several passages the chimeric virus was more infective. Sequence analysis of two separately adapted chimeric viruses showed several mutations. In one of these, one amino acid change in the HCPro is further studied and is shown by mutational analysis to be able to increase the virulence of the chimeric virus.

The manuscript is well written, with correctly performed experiments, described with detail in materials and methods.

questions and comments:
line 66: I suppose the NOT should be deleted
line 218: total RNAs prepared from whole plants or several leaves or only one leaf?

Fig2: passages are not correctly mentioned, since 2nd passage is the 1st for the plantA generations (and 3rd the 2nd).

why was the second (A)/third (D) passage performed on DCL2/4 plants ? and the following on wt and DCL plants?

Fig3: agarose gels part (b) are not necessary

why have chimeras with and withouit 6x-His tag been used in the experiments?

why have the two near mutations in the VPg/NIa cistrons appearing in plant A and D not been studied?

Fig 4: explain in legend what virus p1´and p2´are
Fig 5: I am not sure if part (a) of the fig is necessary.
(b): delete or change the title since polymorphisms are also shown for the 5´NCR (called UTR in the upper part, better unify).
T should be changed into U  
Repeat here, to make it easier for the reader that the mutation studied here is U2053C
line 355: HCPro532? ---next line chimera 353?
line 358: OR DCL2/4 plants
line 369: use "undetectable" instead of "absent"

Fig 6: what are WT1-WT4 ?? why are several lanes not defined in the second blot of the systemic accumulation 10 dai?
legend Fig 6B: relative accumulation determined by RT-qPCR... The RNA extraction from whole plants would possibly reduce the big error bars.

Fig 7: why does the first blot only show the GFP and not CP?

table below:
ND not detectable (or not determined?); why do you differ between NO and ND?

what would happen if in the PVY virus Ile 352 was changed to Thr? Maybe this virus would infect in your hands....?

line 421: our "previous" studies .....
line 430: word "plants" is repeated
line 445: Iwould suggest to delete "on"
line 453: Figure 2b
Do you have an explanation for why the adapted chimeras do not accumulate to higher levels in DCL2/4.5i plants?
line 462: a potential target "to?" viral adaptation. Shouldn´t it be "for"

line 496: libraries "from systemicly" infected tissues.
line 500: mutation OF HCPro amino acid..., we HAVE no indication ....
line 506: Do you have any suggestion why the mutation in HCPro 352 increased the virulence of the chimera? Is it possible that it only leads to incresed viral titers and that this results in movement being detectable earlier, not the virus moving faster?
 line 521: what do you mean with "lacking any initial sequence variability"? that it is generation 1, newly agro-infiltrated?
line 526: leed instead of led
line 528: increase instead of increases

Reviewer 2 Report

Sun et al. investigated host adaptation using PVY-PPV chimeric virus.
The research concepts attracts the plant virologists and the rough design interests the researchers including me.

However, some results are something wired and the information is not enough to evaluate correctly.

You showed the results of western blotting. But I cannot understand which antibody was used in each experiment and these seem no consistency. Only CP was detected from the agroinfiltrated leaves. On the other hand, GFP was detected from tobacco local leaves inoculated virus but both CP and GFP were detected simultaneously from tobacco systemic leaves and Arabidolsis leaves.
These data suggest the possibility of inconsistent experiment condition.
Please explain them.

I cannot find the description of the statistical analysis and you seem performed incorrectly.
For example, it was written in Figure 2 that the quantification of virus titers was evaluated using Tukey HSD, Dunnett T3, Games-Howell, Dunnett t. But these cannot be used simultaneously. 
Figure 6b also contains two methods, Student's T-test and Mann-Whitney. 
You should correctly perform all the statistical analyses and write the analysis condition.
These graphs also lacked the y axis labels and unit.

In the experiment with aphids, the transmission of chimeric virus was failed but it was indicated that the control test with PVY was succeed without any data.
Even if the transmission of chimeric virus wasn't detected, you should show the positive control. If there is no data, you should delete them.

Minor point:

- "HC-Pro" and "VPg" are written better than "HCPro" and "Vpg", respectively. You had better follow ICTV information.
- The description was not unified in the manuscript;
      -"DCL2/4.5i" was sometimes written "DCL2/4". 
      -PCR conditions are written "50 °C for 10 min; 95 °C for 3 min; 40 cycles of 95 °C for 10 s and 60 °C for 20 s; and a final ramp for melting analysis from 60 °C to 95 °C rising 1 °C every 5 s." in L174 but "[98 °C 1 min] 1 cycle; [98 °C 10sc/50 °C 10 sc/72 °C 25 sc] 35 cycles." in L223.

Also I found some typos etc :

L89 as well as not to be italic
L91 "var. Col 0" to "ecotype. Col-0"
L137 "0,3" to "0.3" (same problem is found in fig. 6)
L269 double periods.
L453 figyre to figure
L537 no period? or the paragraph might be not ended?

I recommend you to check the whole manuscript grammatically.

Round 2

Reviewer 2 Report

Thank you for your revision and kind comments.

My concerns have been addressed.